# CNNpack: Packing Convolutional Neural Networks in the Frequency Domain

**Yunhe Wang**[1,3]**, Chang Xu**[2]**, Shan You**[1,3]**, Dacheng Tao**[2]**, Chao Xu**[1,3]
[1]Key Laboratory of Machine Perception (MOE), School of EECS, Peking University
[2]Centre for Quantum Computation and Intelligent Systems,
School of Software, University of Technology Sydney
[3]Cooperative Medianet Innovation Center, Peking University
wangyunhe@pku.edu.cn, Chang.Xu@uts.edu.au, youshan@pku.edu.cn,
Dacheng.Tao@uts.edu.au, xuchao@cis.pku.edu.cn

## Abstract

Deep convolutional neural networks (CNNs) are successfully used in a number of applications. However, their storage and computational requirements have largely prevented their widespread use on mobile devices. Here we present an effective CNN compression approach in the frequency domain, which focuses not only on smaller weights but on all the weights and their underlying connections. By treating convolutional filters as images, we decompose their representations in the frequency domain as common parts (*i.e.*, cluster centers) shared by other similar filters and their individual private parts (*i.e.*, individual residuals). A large number of low-energy frequency coefficients in both parts can be discarded to produce high compression without significantly compromising accuracy. We relax the computational burden of convolution operations in CNNs by linearly combining the convolution responses of discrete cosine transform (DCT) bases. The compression and speed-up ratios of the proposed algorithm are thoroughly analyzed and evaluated on benchmark image datasets to demonstrate its superiority over state-of-the-art methods.

## 1 Introduction

Thanks to the large amount of accessible training data and computational power of GPUs, deep learning models, especially convolutional neural networks (CNNs), have been successfully applied to various computer vision (CV) applications such as image classification [19], human face verification [20], object recognition, and object detection [7, 17]. However, most of the widely used CNNs can only be used on desktop PCs or even workstations due to their demanding storage and computational resource requirements. For example, over 232MB of memory and over $7.24 \times 10^8$ multiplications are required to launch AlexNet and VGG-Net per image, preventing them from being used in mobile terminal apps on smartphones or tablet PCs. Nevertheless, CV applications are growing in importance for mobile device use and there is, therefore, an imperative to develop and use CNNs for this purpose.

Considering the lack of GPU support and the limited storage and CPU performance of mainstream mobile devices, compressing and accelerating CNNs is essential. Although CNNs can have millions of neurons and weights, recent research [9] has highlighted that over 85% of weights are useless and can be set to 0 without an obvious deterioration in performance. This suggests that the gap in demands made by large CNNs and the limited resources offered by mobile devices may be bridged.

Some effective algorithms have been developed to tackle this challenging problem. [8] utilized vector quantization to allow similar connections to share the same cluster center. [6] showed that the weight matrices can be reduced by low-rank decomposition approaches.[4] proposed a network architecture

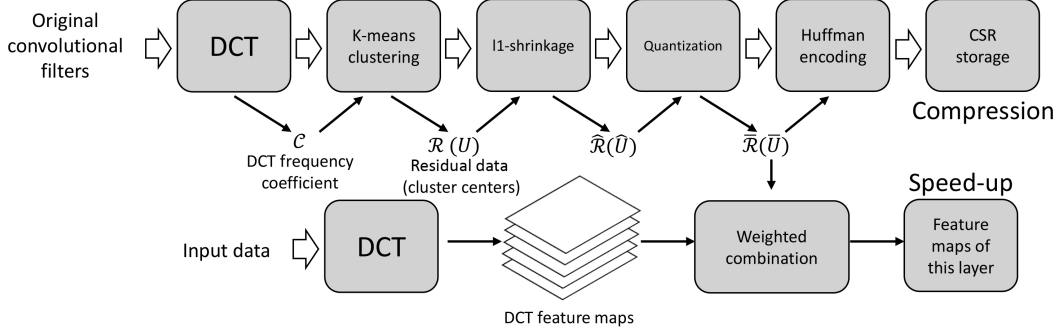

Figure 1: The flowchart of the proposed CNNpack.

using the "hashing trick" and [4] then transferred the HashedNet into the discrete cosine transform (DCT) frequency domain [3]. [16, 5] proposed binaryNet, whose weights were -1/1 or -1/0/1 [2]. [15] utilizes a sparse decomposition to reduce the redundancy of weights and computational complexity of CNNs. [9] employed pruning [10], quantization, and Huffman coding to obtain a greater than $35\times$ compression ratio and $3\times$ speed improvement, thereby producing state-of-the-art CNNs compression to the best of our knowledge. The effectiveness of the pruning strategy relies on the principle that if the absolute value of a weight in a CNN is sufficiently small, its influence on the output is often negligible. However, these methods tend to disregard the properties of larger weights, which might also provide opportunities for compression. Moreover, independently considering each weight ignores the contextual information of other weights.

To address the aforementioned problems, we propose handling convolutional filters in the frequency domain using DCT (see Fig. 1). In practice, convolutional filters can be regarded as small and smooth image patches. Hence, any operation on the convolutional filter frequency coefficients in the frequency domain is equivalent to an operation performed simultaneously over all weights of the convolutional filters in the spatial domain. We factorize the representation of the convolutional filter in the frequency domain as the composition of common parts shared with other similar filters and its private part describing some unique information. Both parts can be significantly compressed by discarding a large number of subtle frequency coefficients. Furthermore, we develop an extremely fast convolution calculation scheme that exploits the relationship between the feature maps of DCT bases and frequency coefficients. We have theoretically discussed the compression and the speed-up of the proposed algorithm. Experimental results on benchmark datasets demonstrate that our proposed algorithm can consistently outperform state-of-the-art competitors, with higher compression ratios and speed gains.

## 2 Compressing CNNs in the Frequency Domain

Recently developed CNNs contain a large number of convolutional filters. We regard convolutional filters as small images with intrinsic patterns, and present an approach to compress CNNs in the frequency domain with the help of the DCT.

### 2.1 The Discrete Cosine Transform (DCT)

The DCT plays an important role in JPEG compression [22], which is regarded as an approximate KL-transformation for 2D images [1]. In JPEGs, the original image is usually divided into several square patches. For an image patch $P \in \mathbb{R}^{n \times n}$, its DCT coefficient $\mathcal{C} \in \mathbb{R}^{n \times n}$ in the frequency domain is defined as:

$$\mathcal{C}_{j_1 j_2} = \mathfrak{D}(P_{i_1 i_2}) = s_{j_1} s_{j_2} \sum_{i_1=0}^{n-1} \sum_{i_2=0}^{n-1} \alpha(i_1, i_2, j_1, j_2) P_{i_1 i_2} = c_{j_1}^T P c_{j_2}, \tag{1}$$

where $s_j = \sqrt{1/n}$ if $j = 0$ and $s_j = \sqrt{2/n}$, otherwise, and $\mathcal{C} = C^T P C$ is the matrix form of the DCT, where $C = [c_1, ..., c_d] \in \mathbb{R}^{d \times d}$ is the transformation matrix. The basis of this DCT is

$S_{j_1 j_2} = c_{j_1} c_{j_2}^T$, and $\alpha(i_1, i_2, j_1, j_2)$ denotes the cosine basis function:

$$\alpha(i_1, i_2, j_1, j_2) = \cos\left(\frac{\pi(2i_1 + 1)j_1}{2n}\right) \cos\left(\frac{\pi(2i_2 + 1)j_2}{2n}\right), \tag{2}$$

and $c_j(i) = \left(\frac{\pi(2i+1)j}{2n}\right)$. The DCT is a lossless transformation thus we can recover the original image by simply utilizing the inverse DCT, *i.e.*,

$$P_{i_1 i_2} = \mathfrak{D}^{-1}(\mathcal{C}_{j_1 j_2}) = \sum_{j_1=0}^{n-1} \sum_{j_2=0}^{n-1} s_{j_1} s_{j_2} \alpha(i_1, i_2, j_1, j_2) \mathcal{C}_{j_1 j_2}, \tag{3}$$

whose matrix form is $P = C C C^T$. Furthermore, to facilitate the notations we denote the DCT and the inverse DCT for vectors as $vec(\mathcal{C}) = \mathfrak{D}(vec(P)) = (C \otimes C) vec(P)$ and $vec(P) = \mathfrak{D}^{-1}(vec(\mathcal{C})) = (C \otimes C)^T vec(\mathcal{C})$, where $vec(\cdot)$ is the vectorization operation and $\otimes$ is the Kronecker product.

## 2.2 Convolutional Layer Compression

**Computing Residuals in the Frequency Domain.** For a given convolutional layer $\mathcal{L}_i$, we first extract its convolutional filters $F^{(i)} = \{F_1^{(i)}, ..., F_{N_i}^{(i)}\}$, where the size of each convolutional filter is $d_i \times d_i$ and $N_i$ is the number of filters in $\mathcal{L}_i$. Each filter can then be transformed into a vector, and together they form a matrix $X_i = [x_1^{(i)}, ..., x_{N_i}^{(i)}] \in \mathbb{R}^{d_i^2 \times N_i}$, where $x_j^{(i)} = vec(F_j^{(i)}), \forall\, j = 1, ..., N_i$.

DCT has been widely used for image compression, since DCT coefficients present an experienced distribution in the frequency domain. Energies of high-frequency coefficients are usually much smaller than those of low-frequency coefficients for 2D natural images, *i.e.*, the high frequencies tend to have values equal or close to zero [22]. Hence, we propose to transfer $X_i$ into the DCT frequency domain and obtain its frequency representation $\mathcal{C} = \mathfrak{D}(X_i) = [\mathcal{C}_1, ..., \mathcal{C}_{N_i}]$. Since a number of convolutional filters will share some similar components, we divide them into several groups $\mathbf{G} = \{\mathbf{G}_1, ..., \mathbf{G}_K\}$ by exploiting $K$ centers $U = [\mu_1, ..., \mu_k] \in \mathbb{R}^{d_i \times K}$ with the following minimization problem:

$$\arg\min_{\mathbf{G}} \sum_{k=1}^{K} \sum_{\mathcal{C} \in \mathbf{G}_k} ||\mathcal{C} - \mu_k||_2^2, \tag{4}$$

where $\mu_k$ is the cluster center of $\mathbf{G}_k$. Fcn. 4 can easily be solved with the conventional k-means algorithm [9, 8]. For each $\mathcal{C}_j$, we denote its residual with its corresponding cluster as $\mathcal{R}_j = \mathcal{C}_j - \mu_{k_j}$, where $k_j = \arg\min_k ||\mathcal{C}_j - \mu_k||_2$. Hence, each convolutional filter is represented by its corresponding cluster center shared by other similar filters and its private part $\mathcal{R}_j$ in the frequency domain. We further employ the following $\ell_1$-penalized optimization problem to control redundancy in the private parts for each convolutional filter:

$$\arg\min_{\widehat{\mathcal{R}}_j} ||\widehat{\mathcal{R}}_j - \mathcal{R}_j||_2^2 + \lambda ||\widehat{\mathcal{R}}_j||_1, \tag{5}$$

where $\lambda$ is a parameter for balancing the reconstruction error and sparsity penalty. The solution to Fcn. 5 is:

$$\widehat{\mathcal{R}}_j = \text{sign}(\mathcal{R}_j) \odot \max\{|\mathcal{R}_j| - \frac{\lambda}{2}, 0\}, \tag{6}$$

where $\text{sign}(\cdot)$ is the sign function. We can also control the redundancy in the cluster centers using the above approach as well.

**Quantization, Encoding, and Fine-tuning.** The sparse data obtained through Fcn. 6 is continuous, which is not benefit for storing and compressing. Hence we need to represent similar values with a common value, *e.g.*, $[0.101, 0.100, 0.102, 0.099] \rightarrow 0.100$. Inspired by the conventional JPEG algorithm, we use the following function to quantize $\widehat{\mathcal{R}}_j$:

$$\overline{\mathcal{R}}_j = \mathcal{Q}\left(\widehat{\mathcal{R}}_j, \Omega, b\right) = \Omega \cdot \mathcal{I}\left\{\frac{\text{Clip}(\widehat{\mathcal{R}}_j, -b, b)}{\Omega}\right\}, \tag{7}$$

where $\text{Clip}(x, -b, b) = \max(-b, \min(b, x))$ with boundary $b > 0$, and $\Omega$ is a large integer with similar functionality to the quantization table in the JPEG algorithm. It is useful to note that the

quantized values produced by applying Fcn. 7 can be regarded as the one-dimensional k-means centers in [9], thus a dictionary can consist of unique values in all $\{\overline{\mathcal{R}}_1, ..., \overline{\mathcal{R}}_{N_i}\}$. Since occurrence probabilities of elements in the codebook are unbalanced, we employ the Huffman encoding for a more compact storage. Finally, the Huffman encoded data is stored in the compressed sparse row format (CSR), denoted as $\mathbf{E}_i$.

It has also been shown that fine-tuning after compression further enhances network accuracy [10, 9]. In our algorithm, we also employ the fine-tuning approach by holding weights that have been discarded so that the fine-tuning operation does not change the compression ratio. After generating a new model, we apply Fcn. 7 again to quantize the new model's parameters until convergence.

The above scheme for compressing convolutional CNNs stores four types of data: the compressed data $\mathbf{E}_i$, the Huffman dictionary with $H_i$ quantized values, the k-means centers $U \in \mathbb{R}^{d_i^2 \times K}$, and the indexes for mapping residual data to centers. It is obvious that if all filters from every layer can share the same Huffman dictionary and cluster centers, the compression ratio will be significantly increased.

### 2.3 CNNpack for CNN Compression

**Global Compression Scheme.** To enable all convolutional filters to share the same cluster centers $U \in \mathbb{R}^{d_i^2 \times k}$ in the frequency domain, we must convert them into a fixed-dimensional space. It is intuitive to directly resize all convolutional filters into matrices of the same dimensions and then apply k-means. However, this simple resizing method increases the amount of the data that needs to be stored. Considering $\overline{d}$ as the target dimension and $d_i \times d_i$ as the convolutional filter size of the $i$-th layer, the weight matrix would be inaccurately reshaped in the case of $d_i < \overline{d}$ or $d_i > \overline{d}$.

A more reasonable approach would be to resize the DCT coefficient matrices of convolutional filters in the frequency domain, because *high-frequency* coefficients are generally small and discarding them only has a small impact on the result $(d_i > \overline{d})$. On the other hand, the introduced zeros will be immediately compressed by CSR since we do not need to encode or store them $(d_i < \overline{d})$. Formally, the resizing operation for convolutional filters in the DCT frequency domain can be defined as:

$$\widehat{\mathcal{C}}_{j_1,j_2} = \Gamma(\mathcal{C}, \overline{d}) = \left\{ \begin{array}{ll} \mathcal{C}_{j_1,j_2}, & \text{if } j_1, j_2 \leq \overline{d}, \\ 0, & \text{otherwise.} \end{array} \right. \tag{8}$$

where $\overline{d} \times \overline{d}$ is the fixed filter size, $\mathcal{C} \in \mathbb{R}^{d_i \times d_i}$ is the DCT coefficient matrix of a filter in the $i$-th layer and $\widehat{\mathcal{C}} \in \mathbb{R}^{\overline{d} \times \overline{d}}$ is the coefficient matrix after resizing. After applying Fcn. 8, we can pack all the coefficient matrices together and use only one set of cluster centers to compute the residual data and then compress the network. We extend the individual compression scheme into an integrated method that is convenient and effective for compressing deep CNNs, which we call CNNpack. The procedures of the proposed algorithm are detailed in the supplementary materials.

CNNpack has five hyper-parameters: $\lambda$, $\overline{d}$, $K$, $b$, and $\Omega$. Its compression ratio can be calculated by:

$$r_c = \frac{\sum_{i=1}^{p} 32 N_i d_i^2}{\sum_{i=1}^{p} \left( N_i \log K + B_i \right) + 32 H + 32 \overline{d}^2 K}, \tag{9}$$

where $p$ is the number of convolutional layers and $H$ is the bits for storing the Huffman dictionary. It is instructive to note that a larger $\lambda$ (Fcn. 5) puts more emphasis on the common parts of convolutional filters, which leads to a higher compression ratio $r_c$. A decrease in any of $b$, $\overline{d}$ and $\Omega$ will increase $r_c$ accordingly. Parameter $K$ is related to the sparseness of $\mathbf{E}_i$, and a larger $K$ would contribute more to $\mathbf{E}_i$'s sparseness but would lead to higher storage requirement. A detailed investigation of all these parameters is presented in Section. 4, and we also demonstrate and validate the trade-off between the compression ratio and CNN accuracy of the convolutional neural work (*i.e.*, classification accuracy [19]).

## 3 Speeding Up Convolutions

According to Fcn. 9, we can obtain a good compression ratio by converting the convolutional filters into the DCT frequency domain and representing them using their corresponding cluster centers and

residual data $\mathbf{R}_i = [\overline{\mathcal{R}}_1, ..., \overline{\mathcal{R}}_{N_i}]$. This is an effective strategy to reduce CNN storage requirements and transmission consumption. However, two other important issues need to be emphasized: memory usage and complexity.

This section focuses on a single layer throughout, we will drop the layer index $i$ in order to simplify notation. In practice, the residual data $\mathbf{R}$ in the frequency domain cannot be simply used to calculate convolutions, so we must first transform them to the original filters in the spatial domain, thus not saving memory. Furthermore, the transformation will further increase algorithm complexity. It is thus necessary to explore a scheme that enables the proposed compression method to easily calculate feature maps of original filters in the DCT frequency domain.

Given a convolutional layer $\mathcal{L}$ with its filters $F = \{F_q\}_{q=1}^N$ of size $d \times d$, we denote the input data as $X \in \mathbb{R}^{H \times W}$ and its output feature maps as $Y = \{Y_1, Y_2, ..., Y_N\}$ with size $H' \times W'$, where $Y_q = F_q * X$ for the convolution operation $*$. Here we propose a scheme which decomposes the calculation of the conventional convolutions in the DCT frequency domain.

For the DCT matrix $C = [c_1, ..., c_d]$, the $d \times d$ convolutional filter $F_q$ can be represented by its DCT coefficient matrix $\mathcal{C}^{(q)}$ with DCT bases $\{S_{j_1,j_2}\}_{j_1,j_2=1}^d$ defined as $S_{j_1,j_2} = c_{j_1} c_{j_2}^T$, namely, $F_q = \sum_{j_1=1}^d \sum_{j_2=1}^d \mathcal{C}_{j_1,j_2}^{(q)} S_{j_1,j_2}$. In this way, feature maps of $X$ through $F$ can be calculated as $Y_q = \sum_{j_1,j_2=1}^d \mathcal{C}_{j_1,j_2}^{(q)} (S_{j_1,j_2} * X)$, where $M = d^2$ is the number of DCT bases. Fortunately, since the DCT is an orthogonal transformation, all of its bases are rank-1 matrices. Note that $S_{j_1,j_2} * X = (c_{j_1} c_{j_2}^T) * X$, and thus feature map $Y_q$ can be written as

$$Y_q = F_q * X = \sum_{j_1,j_2=1}^d \mathcal{C}_{j_1,j_2}^{(q)} (S_{j_1,j_2} * X) = \sum_{j_1,j_2=1}^d \mathcal{C}_{j_1,j_2}^{(q)} [c_{j_1} * (c_{j_2}^T * X)]. \qquad (10)$$

One drawback of this scheme is that when the number of filters in this layer is relatively small, *i.e.*, $M \approx N$, Fcn. 10 will increase computational cost. Moreover, the complexity can be further reduced given the fascinating fact that the feature maps calculated by these DCT bases are exactly the same as the DCT frequency coefficients of the input data. For a $d \times d$ matrix $X$, we consider the matrix form of its DCT, *i.e.*, $\mathcal{C} = C^T X C$, and DCT bases $\{S_{j_1,j_2}\}$, then the coefficient $\mathcal{C}_{j_1,j_2}$ can be calculated as

$$\mathcal{C}_{j_1,j_2} = c_{j_1}^T X c_{j_2} = c_{j_1}^T (c_{j_2}^T * X) = c_{j_1} * (c_{j_2}^T * X) = (c_{j_1} * c_{j_2}^T) * X = (c_{j_1} c_{j_2}^T) * X = S_{j_1,j_2} * X, \text{ (11)}$$

thus we can obtain the feature maps of $M$ DCT bases by only applying the DCT once. Thus the computational complexity of our proposed scheme can be analyzed in Proposition 1.

**Proposition 1.** *Given a convolutional layers with $N$ filters, denote $M = d \times d$ as the number of DCT base, and $\mathcal{C} \in \mathbb{R}^{d^2 \times N}$ as the compressed coefficients of filters in this layer. Suppose $\delta$ is the ratio of non-zero elements in $\mathcal{C}$, while $\eta$ is the ratio of non-zero elements in $K'$ active cluster centers of this layer. The computational complexity of our proposed scheme is $\mathcal{O}((d^2 \log d + \eta M K' + \delta M N) H' W')$.*

The proof of Proposition 1 can be found in the supplementary materials. According to Proposition 1, the proposed compression scheme in the frequency domain can also improve the speed. Compared to the original CNN, for a convolutional layer, the speed-up of the proposed method is

$$r_s = \frac{d^2 N H' W'}{(d^2 \log d + \eta K' M + \delta N M) H' W'} \approx \frac{N}{\eta K' + \delta N}. \qquad (12)$$

Obviously, the speed-up ratio of the proposed method is directly relevant to $\eta$ and $\delta$, which correspond to $\lambda$ in Fcn. 6.

## 4   Experimental Results

**Baselines and Models.** We compared the proposed approach with 4 baseline approaches: Pruning [10], P+QH (Pruning + Quantization and Huffman encoding) [9], SVD [6], and XNOR-Net [16]. The evaluation was conducted using the MNIST and ILSVRC2012 datasets. We compared the proposed compression approach with four baseline CNNs: LeNet [14, 21], AlexNet [13], VGG-16 Net [19], and ResNet-50 [11]. All methods were implemented using MatConvNet [21] and run on

NVIDIA Titan X graphics cards. Model parameters were stored and updated as 32 bit floating-point values.

**Impact of parameters.** As discussed above, the proposed compression method has several important parameters: $\lambda$, $\bar{d}$, $K$, $b$, and $\Omega$. We first tested their impact on the network accuracy by conducting an experiment using MNIST [21], where the network has two convolutional layers and two fully-connected layers of size $5 \times 5 \times 1 \times 20$, $5 \times 5 \times 20 \times 50$, $4 \times 4 \times 50 \times 500$, and $1 \times 1 \times 500 \times 10$, respectively. The model accuracy was 99.06%. The compression results of different $\lambda$ and $\bar{d}$ after fine-tuning 20 epochs are shown in Fig. 2 in which $k$ was set as 16, $b$ was equal to $+\infty$ since it did not make an obvious contribution to the compression ratio even when set at a relatively smaller value (*e.g.*, $b = 0.05$) but caused the accuracy reduction. $\Omega$ was set to 500, making the average length of weights in the frequency domain about 5, a bit larger than that in [9] but more flexible and with relatively better performance. Note that all the training parameters used their the default settings, such as for epochs, learning rates, *etc.*.

It can be seen from Fig. 2 that although a lower $\bar{d}$ slightly improves the compression ratio and speed-up ratio simultaneously, this comes at a cost of decreased overall network accuracy; thus, we kept $\bar{d} = \max\{d_i\}$, $\forall\, i = 1, ..., p$, in CNNpack. Overall, $\lambda$ is clearly the most important parameter in the proposed scheme, which is sensitive but monotonous. Thus, it only needs to be adjusted according to demand and restrictions. Furthermore, we tested the impact of number of cluster centers $K$. As mentioned above, $K$ is special in that its impact on performance is not intuitive, and when it becomes larger, $\mathbf{E}$ becomes sparser but needs more space for storing cluster centers $U$ and indexes. Fig. 3 shows that $K = 16$ provides the best trade-off between compression performance and accuracy.

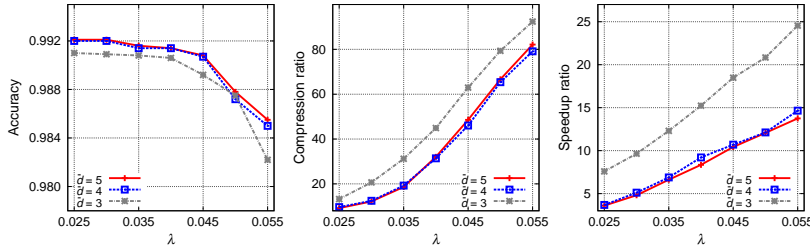

Figure 2: The performance of the proposed approach with different $\lambda$ and $\bar{d}$.

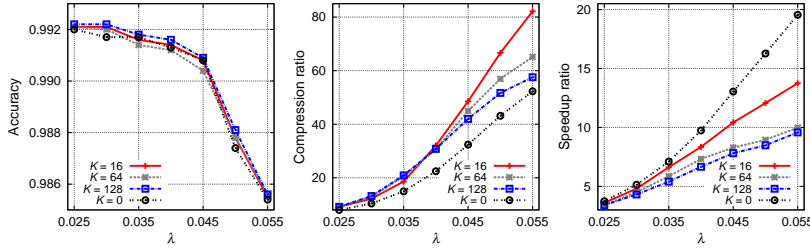

Figure 3: The performance of the proposed approach with different numbers of cluster centers $K$.

We also report the compression results by directly compressing the DCT frequency coefficients of original filters $\mathcal{C}$ as before (*i.e.*, $K = 0$, the black line in Fig. 3). It can be seen that the clustering number does not affect accuracy, but a suitable $K$ does enhance the compression ratio. Another interesting phenomenon is that the speed-up ratio without decomposition is larger than that of the proposed scheme because the network is extremely small and the clustering introduces additional computational cost as shown in Fcn. 12. However, recent networks contain a lot more filters in a convolutional layer, larger than $K = 16$.

Based on the above analysis, we kept $\lambda = 0.04$ and $K = 16$ for this network (an accuracy of 99.14%). Accordingly, the compression ratio $r_c = 32.05\times$ and speed-up ratio $r_s = 8.34\times$, which is the best trade-off between accuracy and compression performance.

**Filter visualization.** The proposed algorithm operates in the frequency domain, and although we do not need to invert the compressed net when calculating convolutions, we can reconstruct the convolutional filters in the spatial domain to provide insights into our approach. Reconstructed convolution filters are obtained from the LeNet on MNIST, as shown in Fig. 4.

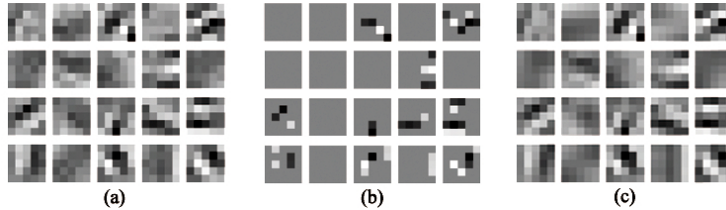

Figure 4: Visualization of example filters learned on MNIST: (a) the original convolutional filters, (b) filters after pruning, (c) convolutional filters compressed by the proposed algorithm.

The proposed approach is fundamentally different to the previously used pruning algorithm. According to Fig. 4(b), weights with smaller magnitudes are pruned while influences of larger weights have been discarded. In contrast, our proposed algorithm not only handles the smaller weights but also considers impacts of those larger weights. Most importantly, we accomplish the compressing task by exploring the underlying connections between all the weights in the convolutional filter (see Fig. 4(c)).

**Compression AlexNet and VGGNet on ImageNet.** We next employed CNNpack for CNN compression on the ImageNet ILSVRC-2012 dataset [18], which contains over 1.2M training images and 50k validation images. First, we examined two conventional models: AlexNet [13], with over 61M parameters and a top-5 accuracy of 80.8%; and VGG-16 Net, which is much larger than the AlexNet with over 138M parameters and has a top-5 accuracy of 90.1%. Table 1 shows detailed compression and speed-up ratios of the AlexNet with $\lambda = 0.04$ and $K = 16$. The result of the VGG-16 Net can be found in the supplementary materials. The reported multiplications are for computing one image.

Table 1: Compression statistics for AlexNet.

| Layer | Num of Weights | Memory | $r_c$ | Multiplication | $r_s$ |
|-------|----------------|--------|-------|----------------|-------|
| conv1 | $11 \times 11 \times 3 \times 96$ | 0.13MB | 878× | $1.05 \times 10^8$ | 127× |
| conv2 | $5 \times 5 \times 48 \times 256$ | 1.17MB | 94× | $2.23 \times 10^8$ | 28× |
| conv3 | $3 \times 3 \times 256 \times 384$ | 3.37MB | 568× | $1.49 \times 10^8$ | 33× |
| conv4 | $3 \times 3 \times 192 \times 384$ | 2.53MB | 42× | $1.12 \times 10^8$ | 15× |
| conv5 | $3 \times 3 \times 192 \times 256$ | 1.68MB | 43× | $0.74 \times 10^8$ | 12× |
| fc6 | $6 \times 6 \times 256 \times 4096$ | 144MB | 148× | $0.37 \times 10^8$ | 100× |
| fc7 | $1 \times 1 \times 4096 \times 4096$ | 64MB | 15× | $0.16 \times 10^8$ | 8× |
| fc8 | $1 \times 1 \times 4096 \times 1000$ | 15.62MB | 121× | $0.04 \times 10^8$ | 60× |
| Total | 60954656 | 232.52MB | 39× | $7.24 \times 10^8$ | 25× |

We achieved a 39× compression ratio and a 46× compression ratio for AlexNet and VGG-16 Net, respectively. The layer with a relatively larger filter size had a larger compression ratio because it contains more subtle high-frequency coefficients. In contrast, the highest speed-up ratio was often obtained on the layer whose filter number $N$ was much larger than its filter size, *e.g.*, the fc6 layer of AlexNet. We obtained an about 9× speed-up ratio on VGG-16 Net, which is lower than that on the AlexNet since complexity is relevant to the feature map size and the first several layers definitely have more multiplications. Unfortunately, their filter numbers are relatively small and their compression ratios are all small, thus the overall speed-up ratio is lower than that on AlexNet. Accordingly, when we set $K = 0$, the compression ratio and the speed-up ratio of AlexNet were close to 35× and 22× and those of VGG-16 Net were near 28× and 7×. This reduction is due to these two networks being relatively large and containing many similar filters. Moreover, the filter number in each layer is larger than the number of cluster centers, *i.e.*, $N > K$. Thus, cluster centers can effectively reduce memory consumption and computational complexity simultaneously.

**ResNet-50 on ImageNet.** Here we discuss a more recent work, ResNet-50 [11], which has more than 150 layers and 54 convolutional layers. This model achieves a top-5 accuracy of 7.71% and a top-1 accuracy of 24.62% with only about 95MB parameters [21]. Moreover, since most of its convolutional filters are $1 \times 1$, *i.e.*, its architecture is very thin with less redundant weights, it is very hard to perform compression methods on it.

For the experiment on ResNet-50, we set $K = 0$ since the functionality of Fcn. 7 for 1-dimensional filters are similar to that of k-means clustering, thus cluster centers are dispensable for this model. Further, we set $\lambda = 0.04$ and therefore discarded about 84% of original weights in the frequency

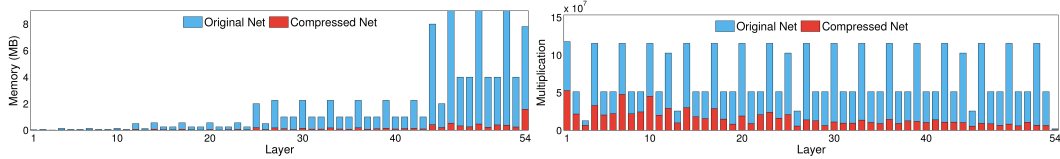

(a) Compression ratios of all convolutional layers. (b) Speed-up ratios of all convolutional layers.

Figure 5: Compression statistics for ResNet-50 (better viewed in color version).

domain. After fine-tuning, we obtained a 7.82% top-5 accuracy on ILSVRC2012 dataset. Fig. 5 shows the detailed compression statistics of ResNet-50 utilizing the proposed CNNpack. In summary, memory usage for storing its filters was squeezed by a factor of $12.28\times$.

It is worth mentioning that, larger filters seem have more redundant weights and connections because compression ratios on layers with $1\times1$ filters are smaller than those on layers with $3\times3$ filters. On the other side, these $1\times1$ filters hold a larger proportion of multiplications of the ResNet, thus we got an about $4\times$ speed-up on this network.

**Comparison with state-of-the-art methods.** We detail a comparison with state-of-the-art-methods for compressing DNNs in Table 2. CNNpack clearly achieves the best performance in terms of both the compression ratio ($r_c$) and the speed-up ratio ($r_s$). Note that although Pruning+QH achieves a similar compression ratio to the proposed method, filters in their algorithm are stored after applying a modified CSC format which stores index differences, means that it needs to be decoded before any calculation. Hence, the compression ratio of P+QH will be lower than that have been reported in [9] if we only consider memory usage. In contrast, the compressed data produced by our method can be directly used for network calculation. In reality, online memory usage is the real restriction for mobile devices, and the proposed method is superior to previous works in terms of both the compression ratio and the speed-up ratio.

Table 2: An overall comparison of state-of-the-art methods for deep neural network compression and speed-up, where $r_c$ is the compression ratio and $r_s$ is the speed-up.

| Model | Evaluation | Original | Pruning [10] | P+QH [9] | SVD [6] | XNOR [16] | CNNpack |
|---|---|---|---|---|---|---|---|
| AlexNet | $r_c$ | 1 | $9\times$ | $35\times$ | $5\times$ | $64\times$ | $39\times$ |
|  | $r_s$ | 1 | - | - | $2\times$ | $58\times$ | $25\times$ |
|  | $top\text{-}1\ err$ | 41.8% | 42.7% | 42.7% | 44.0% | 56.8% | 41.6% |
|  | $top\text{-}5\ err$ | 19.2% | 19.6% | 19.7% | 20.5% | 31.8% | 19.2% |
| VGG16 | $r_c$ | 1 | $13\times$ | $49\times$ | - | - | $46\times$ |
|  | $r_s$ | 1 | - | $3.5\times$ | - | - | $9.4\times$ |
|  | $top\text{-}1\ err$ | 28.5% | 31.3 | 31.1% | - | - | 29.7% |
|  | $top\text{-}5\ err$ | 9.9% | 10.8 | 10.9% | - | - | 10.4% |

# 5 Conclusion

Neural network compression techniques are desirable so that CNNs can be used on mobile devices. Therefore, here we present an effective compression scheme in the DCT frequency domain, namely, CNNpack. Compared to state-of-the-art methods, we tackle this issue in the frequency domain, which can offer the probability for more compression ratio and speed-up. Moreover, we no longer independently consider each weight since each frequency coefficient's calculation involves all weights in the spatial domain. Following the proposed compression approach, we explore a much cheaper convolution calculation based on the sparsity of the compressed net in the frequency domain. Moreover, although the compressed network produced by our approach is sparse in the frequency domain, the compressed model has the same functionality as the original network since filters in the spatial domain have preserved intrinsic structure.Our experiments show that the compression ratio and the speed-up ratio are both higher than those of state-of-the-art methods. The proposed CNNpack approach creates a bridge to link traditional signal and image compression with CNN compression theory, allowing us to further explore CNN approaches in the frequency domain.

**Acknowledgements** This work was supported by the National Natural Science Foundation of China under Grant NSFC 61375026 and 2015BAF15B00, and Australian Research Council Projects: FT-130101457, DP-140102164 and LE-140100061.

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
