[Supplementary Material]

# CNNpack: Packing Convolutional Neural Networks in the Frequency Domain (Supplementary Materials)

**Yunhe Wang**[1,3]**, Chang Xu**[2]**, Shan You**[1,3]**, Dacheng Tao**[2]**, Chao Xu**[1,3]
[1]Key Laboratory of Machine Perception (MOE), School of EECS, Peking University
[2]Centre for Quantum Computation and Intelligent Systems,
School of Software, University of Technology Sydney
[3]Cooperative Medianet Innovation Center, Peking University
wangyunhe@pku.edu.cn, Chang.Xu@uts.edu.au, youshan@pku.edu.cn,
Dacheng.Tao@uts.edu.au, xuchao@cis.pu.edu.cn

In this document we give the proof of the Proposition 1 in main body, the algorithm of the proposed compression algorithm in Section 2.3. We also report some detailed experimental results *e.g.*, the compression statistics for VGG-16 Net [4].

## 1 Proofs of Proposition 1 in main body

**Proposition 1.** *Given a convolutional layers with $N$ filters, denote $M = d \times d$ as the number of DCT base, and $\mathcal{C} \in \mathbb{R}^{d^2 \times N}$ as the compressed coefficients of filters in this layer. Suppose $\delta$ is the ratio of non-zero elements in $\mathcal{C}$, while $\eta$ is the ratio of non-zero elements in $K'$ active cluster centers of this layer. The computational complexity of our proposed scheme is $\mathcal{O}((d^2 \log d + \eta M K' + \delta M N) H' W')$.*

*Proof.* The computational complexity for the feature maps $Y$ can be computed as $\mathcal{O}(d^2 N H' W')$. When implementing the compressed CNN with our proposed algorithm, a naive approach would be to invert all frequency-filters into the spatial domain and then calculate spatial convolutions. Since the computational complexity of a $d \times d$ DCT is $\mathcal{O}(d^2 \log d)$ [1], the overall complexity of the method will be $\mathcal{O}(d^2 \log dN + d^2 N H' W')$ or, equally, $\mathcal{O}(d^2 N H' W')$ considering $d^2 \log dN \ll d^2 N H' W'$. However, this simple method tends to be inefficient since it involves a lot of redundant computation. Hence, we propose to first use the DCT bases as a set of filter bases to obtain a set of feature maps; the feature map of a convolutional filter can then be quickly calculated by summarizing them based on their DCT coefficients $\mathcal{C}$.

Since we decompose the convolutions by combinations of feature maps of DCT bases in Fcn. 10, the complexity should be rewritten as $\mathcal{O}((2dM + MN)H'W')$. Moreover, the complexity of a $d \times d$ DCT is only $\mathcal{O}(d^2 \log d)$ [1], thus feature maps of those DCT bases can be quickly calculated by using Fcn. 11, thus the complexity of Fcn. 10 is reduced to $\mathcal{O}((d^2 \log d + MN)H'W')$. Furthermore, if $\mathbf{R}$ is sufficiently sparse, the complexity can be rewritten as

$$\mathcal{O}((d^2 \log d + \sum_{i=1}^{N} ||\mathbf{R}||_0)H'W'). \tag{1}$$

It is important to note that the complexity of Fcn. 1 would be significantly smaller than the original complexity given $\sum_{i=1}^{N} ||\mathbf{R}||_0 \ll MN$. We can simplify it as $\sum_{i=1}^{N} ||\mathbf{R}||_0 = \delta MN$, where $\delta$ is a small value (*e.g.*, $\delta = 0.05$) denoting the sparse degree of the compressed CNN. A stronger sparseness penalty would encourage $\delta$ to be smaller.

As well as the residual data $\mathbf{R}$, we must also calculate the filter maps of cluster centers $U$. Since $U$ has been obtained over all convolutional filters in different layers in a network, if the convolutional filters in a layer correspond to only $K'$ centers (where $K' \leq K$), additional computational cost will

be saved in this layer. The complexity can be further decreased if we also apply Fcn. 6 to shrink all cluster centers $U$ before the calculation of residual data $\mathbf{R}$,

$$\mathcal{O}((d^2 \log d + \sum_{k=1}^{K'} ||U'_k||_0)H'W') = \mathcal{O}((d^2 \log d + \eta M K')H'W'), \tag{2}$$

where $\eta$ is similar to $\delta$ and denotes the sparse degree of cluster centers $U'$ for this layer. Since we only need to calculate the feature maps of DCT bases once, the complexity of our proposition is

$$\mathcal{O}((d^2 \log d + \eta M K' + \delta M N)H'W'). \tag{3}$$

$\square$

## 2 Algorithm for the Proposed Compression Scheme

In Section 2.3, we had proposed an algorithm for compressing CNNs, which pack all the convolutional layers together and forms a compact representation in the DCT frequency domain. Alg. 1 summarizes the procedures of the proposed algorithm.

---
**Algorithm 1** CNNpack for compressing deep convolutional neural networks.

---
**Input:** A pre-trained convolutional neural network with $p$ convolutional layers $\mathcal{L}_1, ..., \mathcal{L}_p$. The dimension and parameters of CNNpack: $\overline{d} \times \overline{d}$, $\lambda$, $K$, $b$ and $\Omega$.

1: **Module 1: Filter extraction and transformation.**
2:   **for** each convolutional layers $\mathcal{L}_i$ in the network **do**
3:     **for** each convolutional filter $F_j^{(i)}$ in $\mathcal{L}_i$ **do**
4:       Vectorize $F_j^{(i)}$: $X_i \leftarrow [vec(F_j^{(i)}), ..., vec(F_{N_i}^{(i)})]$;
5:       Transfer $X_i$ into the DCT frequency domain: $\mathcal{C} \leftarrow \mathfrak{D}(X_i)$ (Fcn. 1));
6:       Resize each $\mathcal{C}_j$ in $\mathcal{C}$ to a $\overline{d} \times \overline{d}$ matrix: $\mathcal{C}_j \leftarrow \Gamma(\mathcal{C}_j, \overline{d})$ (Fcn. 8);
7:     **end for**
8:   **end for**
9: **Module 2: Clustering and residual coding.**
10: Employ k-means (Fcn. 4) to generate $K$ cluster centers $U = [\mu_1, ..., \mu_K]$;
11: **for** each convolutional layers $\mathcal{L}_i$ in the network **do**
12:   **for** each column $\mathcal{C}_j$ in $\mathcal{C}$ **do**
13:     Subtract the closest center $u_{j_k}$ ($u_{j_k} \in U$, $s.t.$ $\min ||\mathcal{C}_j - \mu_{j_k}||_2$) of $\mathcal{C}_j$: $\mathcal{R}_j \leftarrow \mathcal{C}_j - \mu_{j_k}$;
14:     Calculate the optimal sparse representation $\widehat{\mathcal{R}}_j$ (Fcn. 6);
15:     Quantize: $\overline{\mathcal{R}}_j \leftarrow \mathcal{Q}\left(\widehat{\mathcal{R}}_j, \Omega, b\right)$;(Fcn. 7);
16:   **end for**
17: **end for**
18: **Module 3: Fine-tuning and compressing.**
19: **repeat**
20:   Fine-tune the compressed network by keeping the discarded components;
21:   **for** each convolutional layers $\mathcal{L}_i$ in the network **do**
22:     Quantize: $\overline{\mathcal{R}}_j \leftarrow \mathcal{Q}\left(\widehat{\mathcal{R}}_j, \Omega, b\right)$;
23:     Compress $\{\overline{\mathcal{R}}_1, ..., \overline{\mathcal{R}}_{N_i}\}$ by exploiting CSR and Huffman encoder to form $\mathbf{E}_i$;
24:   **end for**
25: **until** convergence
**Output:** The compressed data $\{\mathbf{E}_i\}$, Huffman dictionary, cluster centers $U$ and indexes.

---

## 3 More Experimental Results

Here we give the performance of the proposed CNNpack on the VGG-16 Net [4], which is much larger than the AlexNet with over 138M parameters and has a top-5 accuracy of 90.1%. The experiment was conducted on the ILSVRC-2012 dataset [3], detailed compression and speed-up ratios is shown in Table 1 with $\lambda = 0.04$ and $K = 16$.

Table 1: Compression statistics for VGG-16 Net.

| Layer | Num of Weights | Memory | $r_c$ | Multiplication | $r_s$ |
|---|---|---|---|---|---|
| conv1_1 | $3 \times 3 \times 3 \times 64$ | 0.006MB | $275\times$ | $0.11\times10^9$ | $25\times$ |
| conv1_2 | $3 \times 3 \times 64 \times 64$ | 0.14MB | $26\times$ | $2.41\times10^9$ | $7\times$ |
| conv2_1 | $3 \times 3 \times 64 \times 128$ | 0.28MB | $14\times$ | $1.20\times10^9$ | $6\times$ |
| conv2_2 | $3 \times 3 \times 128 \times 128$ | 0.56MB | $14\times$ | $2.41\times10^9$ | $6\times$ |
| conv3_1 | $3 \times 3 \times 128 \times 256$ | 1.12MB | $16\times$ | $1.20\times10^9$ | $8\times$ |
| conv3_2 | $3 \times 3 \times 256 \times 256$ | 2.25MB | $15\times$ | $2.41\times10^9$ | $8\times$ |
| conv3_3 | $3 \times 3 \times 256 \times 256$ | 2.25MB | $28\times$ | $2.41\times10^9$ | $13\times$ |
| conv4_1 | $3 \times 3 \times 256 \times 512$ | 4.5MB | $12\times$ | $1.20\times10^9$ | $8\times$ |
| conv4_2 | $3 \times 3 \times 512 \times 512$ | 9MB | $40\times$ | $2.41\times10^9$ | $21\times$ |
| conv4_3 | $3 \times 3 \times 512 \times 512$ | 9MB | $45\times$ | $2.41\times10^9$ | $23\times$ |
| conv5_1 | $3 \times 3 \times 512 \times 512$ | 9MB | $8\times$ | $0.60\times10^9$ | $5\times$ |
| conv5_2 | $3 \times 3 \times 512 \times 512$ | 9MB | $12\times$ | $0.60\times10^9$ | $8\times$ |
| conv5_3 | $3 \times 3 \times 512 \times 512$ | 9MB | $21\times$ | $0.60\times10^9$ | $12\times$ |
| fc6 | $7 \times 7 \times 512 \times 4096$ | 392MB | $208\times$ | $0.41\times10^9$ | $158\times$ |
| fc7 | $1 \times 1 \times 4096 \times 4096$ | 64MB | $14\times$ | $0.16\times10^8$ | $8\times$ |
| fc8 | $1 \times 1 \times 4096 \times 1000$ | 15.62MB | $215\times$ | $0.41\times10^7$ | $120\times$ |
| Total | 138344128 | 572.74MB | $46\times$ | $2.04 \times 10^{10}$ | $9.4\times$ |

Furthermore, we give the enlarged figures (Fig. 1) of the results of the ResNet-50 [2] in order to have a better illustration.

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

(a) Compression ratios of all convolutional layers. (b) Speed-up ratios of all convolutional layers.

Figure 1: Compression statistics for ResNet-50.