[Reviews · NeurIPS 2016]

Reviewer 1

Summary

The paper compresses convolutional layers of CNNs using K-means clustering in discrete cosine transform space. Quantization and Huffman encoding are applied as in previous work. The use of DCTs provide a nontrivial speedup over previous work, but not significantly more compression (slightly more on AlexNet, slightly less on VGG16).

Qualitative Assessment

The technique is fairly natural, since DCTs are routinely used to compress images. However, neural network weight matrices look quite different from images especially in the case of weight decay. They are often sparse, which lends strength to techniques that take advantage of sparsity elementwise rather than sparsity after transformation. This handwaving is backed up by the paper results, which do not exhibit significantly more compression than the P+QH baseline. Further notes: 1. The comparison to only AlexNet and VGG on ImageNet is problematic for judging compression quality, since Inception is significantly smaller than both models. The possibly underconverged non-augmented version on CIFAR-10 is insufficient here, since a better tuned model might be more difficult to compress without losing accuracy. 2. "It is obvious that if all filters from every layer can share the same Huffman dictionary and cluster centers, the compression ratio will be significantly increased." This is not obvious, so please justify. If the bulk of the cost went towards the residuals and the Hamming encoded bits, and not towards the codebooks, this would not hold. Moreover, varying the Huffman codebook per layer can improve compression arbitrarily on some data sets (consider one layer of all zeros and another layer of all ones). More generally, please provide the breakdown of storage costs across the different components of the format, si nce this is necessary for evaluating the method and its tradeoffs. 3. "monotonous" is the wrong word; please proofread the paper for grammar and typos. 5. Is the P+QH baseline from [9] fine tuned? There appears to be some degradation in accuracy which [9] did not report. Not fine tuning the baseline would be a bad comparison, so if you did fine tune, please say so. 7. The "Comparison with state-of-the-art methods" section says that [9] requires decoding due to Huffman but this method does not. This is confusing, since the new method also uses Huffman encoding.

Confidence in this Review

2-Confident (read it all; understood it all reasonably well)


Reviewer 2

Summary

The paper proposes a new compression and speedup scheme for ConvNets. The scheme involves: - Take DCT of layer filters - Do K-means clustering on DCT coefficients and store cluster centers + residuals - Further compress the coefficients by quantization - Store weights in CSR format They finally figure out how to directly use these DCT coefficients to do the layer-wise compute, and since there's reuse possible here (several filters might share large parts of computation), they get speedups as well. (ignoring mnist experiments) They test on AlexNet, VGG, Googlenetv3 on Imagenet, Cifar-10

Qualitative Assessment

It was hard to follow the paper due to the overuse of notation to describe simple algorithms. The authors should consider adding a more visual explanation (apart from their flow-chart). I am an expert in the area of model compression and it was hard to follow, so imagine the general reader. Overall, the paper has a novel approach of tackling model compression that is interesting, especially because they show how this gives one compute speedups as well. I was enjoying the paper until I came to the nonsense section: "GoogleNet Inception v3 on ImageNet." Remove it if you dont have time to do experiments properly. Your paper is still strong without it, and you do all kinds of bad experiments and claims there.

Confidence in this Review

3-Expert (read the paper in detail, know the area, quite certain of my opinion)


Reviewer 3

Summary

The paper presents a new approach to compressing (reduced memory) and speeding up the computations of CNNs by moving the compression schemes to the frequency domains. It compares the proposed algorithm to the state of the art schemes on benchmark models and datasets and demonstrates significant improvements.

Qualitative Assessment

Good paper. - Idea to move compression to frequency domain is a nice addition to this field - Demonstrates success on a good set of benchmarks agains the state of the art in this area - Significant improvements on existing state of the art Minor con: Would be great to get full results on ImageNet for GoogleNet v3, but it does take a long time. If possible, will be great to get that for the final paper/conference.

Confidence in this Review

3-Expert (read the paper in detail, know the area, quite certain of my opinion)


Reviewer 4

Summary

The paper studies compression of convolutional neural nets. They proposed compression in the frequency domain of convolutional nets combined with clustering of the resulting coefficients. The paper is well written and easy to read. The experiments are well thought and comprehensive.

Qualitative Assessment

I think the idea of using compression in the frequency domain of convolutional layers is novel despite already being used for images (outside of deep learning). Clustering also makes sense. The experiments are comprehensive and well done.

Confidence in this Review

3-Expert (read the paper in detail, know the area, quite certain of my opinion)


Reviewer 5

Summary

This paper proposes a compression algorithm in the frequency domain. The algorithm aims at addressing properties of larger weights and the relation between coefficients within a kernel. The main idea is factorizing filters as a composition of common components and 'independent' ones for each filter. A really nice thing (I was not aware of) is the fact that this is done globally for all the filters in the network (the so call global compact). The algorithm needs several post-processing steps such as the DCT and then minimizing the reconstruction plus the pruning and quantization and ultimately the fine tuning part. Experiments show the improvement in terms of trade-off between accuracy, speed and compression when used on large networks (AlexNet and VGG) based on the parameter setting from small ones (LeNet type). What is not explicitly clear is the performance compared to pure baselines. That is the part that seems to be a bit hidden in the paper.

Qualitative Assessment

I like the paper and the idea of treating all the filters simultaneously. However, after reading several times the introduction it does not seem to be highlighted and importantly, related work is not clear in that sense. I am aware of per layer compression but not global compression. Would be nice to be clearer on this. One potential problem is the selection of K. The paper addresses this problem using MNIST and then transferring results (to some extent). The discussion around compressing and speeding up nets especially the last layers seems weird (lines ~240) as the most time-consuming part is in the first layers and the larger number of parameters is in the fully connected layers. I am a bit concerned about the lack of clarity in the accuracy drop of the experiments. While MNIST experiment clearly shows this, experiments on ImageNet are obscure in that sense. For instance, the performance is separated from compression analysis and performance of Inception is not discussed / shown. I also wonder why using AlexNet and VGG for compression. These networks use a large amount of memory and parameters mainly in fully connected layers. It has been shown that these layers can be removed. In that case, would be better for comparison. The paper main contribution is on the Global compression considering all the filters together (section 2.3). However, it is not clear to me how this is done when the fully connected layer is present and the impact of going from 1x1 convolutions to 11x11 in the case of ImageNet. That is briefly discussed in lines 118 / 119 but nowhere else.

Confidence in this Review

2-Confident (read it all; understood it all reasonably well)


Reviewer 6

Summary

This paper presents a framework of compressing and speeding up convolutional neural networks. The convolutional kernels are transformed into frequency domain with Discrete Cosine Transform. k-means clustering, l1 shrinking, quantization and Huffman coding is performed sequentially to reduce the consumption of memory and computational complexity. Excellent experimental results are achieved on several state of the art CNN models.

Qualitative Assessment

1. The experimental results are exceptionally good, significantly better than previous methods. The authors evaluated their method on Alexnet, VGG and Googlenet, which I think are sufficiently convincing. 2. Concerns about novelty: Comparing to [1], the main added part is the DCT transformation. While speeding up CNN with frequency domain has been proposed in [2]. 3. The speedup convolution section seems confusing to me. 3.1 I believe the DCT mentioned in this paper is 2-dimensional DCT, as is defined in equation (2). The complexity of 2D DCT of a d x d patch should not be O(dlog(d)). I believe O(dlog(d)) is the complexity of 1D DCT with d elements. 3.2 Line 166 to 168 seems to be misleading. The author says that convolving with the d x d DCT bases is equivalent to performing DCT direct to the H x W input data. However, it should be performing d x d DCT at each location of the H x W input data. These are two different operations. 3.3 When calculating complexity, the authors never mentioned the number of input channels. Is the number of input channels supposed to be one? 4. I suggest the authors to compare their method with [3]. DCT is a type of decomposition with fixed basis, while in [3], sparse decomposition was adopted, and the basis is optimized while fine-tuning. 5. I'm interested in the values of eta and delta for the models evaluated in the experiments. eta and delta provides the level of sparsity for the k-means centers and residuals. This can be compared with the sparsity obtained in [3]. 6. The run time of proposed method is given in Table 2 of the manuscript. I'm very curious about how the authors implemented their method with GPU. According to [3], performing sparse matrix multiplication heavily suffers from the limited memory bandwidth, and GPU is well known to be very hard to optimize. Considering the proposed method is relatively complicated, I'm wondering how the authors parallelize it efficiently with CUDA and achieve runtime speedup that is similar to theoretical speedup. I suggest the authors compare their efficiency with the cudnn implementation, which represents the state-of-the-art implementation of CNN models. [1] Han, Song, Huizi Mao, and William J. Dally. "Deep compression: Compressing deep neural network with pruning, trained quantization and huffman coding." CoRR, abs/1510.00149 2 (2015). [2] Mathieu, Michael, Mikael Henaff, and Yann LeCun. "Fast training of convolutional networks through FFTs." arXiv preprint arXiv:1312.5851 (2013). [3] Liu, Baoyuan, et al. "Sparse convolutional neural networks." Proceedings of the IEEE Conference on Computer Vision and Pattern Recognition. 2015.

Confidence in this Review

3-Expert (read the paper in detail, know the area, quite certain of my opinion)